# SplitRegex: Faster Regex Synthesis Framework via Neural Example Splitting

## Abstract

Due to the practical importance of regular expressions (regexes, for short), there has been a lot of research to automatically generate regexes from positive and negative string examples. A basic idea of learning a regex is a search-and-repair; search for a correct regex and repair it if incorrect. The problem is known to be PSPACE-complete and the main issue is to obtain a regex quickly within a time limit. While classical regex learning methods do not perform well, recent approaches using deep neural networks show better performance with respect to the accuracy of the resulting regexes. However, all these approaches including SOTA models are often extremely slow because of the slow searching mechanism, and do not produce desired regexes within a given time limit.

We tackle the problem of learning regexes faster from positive and negative strings by relying on a novel approach called 'neural example splitting'. Our approach essentially split up example strings into multiple parts using a neural network trained to group similar substrings from positive strings. This helps to learn a regex faster and, thus, more accurately since we now learn from several short-length strings. We propose an effective regex synthesis framework called 'SplitRegex' that synthesizes subregexes from 'split' positive substrings and produces the final regex by concatenating the synthesized subregexes. For the negative sample, we exploit pre-generated subregexes during the subregex synthesis process and perform the matching against negative strings. Then the final regex becomes consistent with all negative strings. SplitRegex is a divided-and-conquer framework for learning target regexes; split (=divide) positive strings and infer partial regexes for multiple parts, which is much more accurate than the whole string inferring, and concatenate (=conquer) inferred regexes while satisfying negative strings. We empirically demonstrate that the proposed SplitRegex framework substantially improves the previous regex synthesis approaches over four benchmark datasets.

## 1 Introduction

Regular expressions (regexes in short) are a powerful tool on processing sequence data in NLP or information retrieval. A regex—a finite sequence—is a formal representation of (infinite) symbol sequences. However, it is not easy to write a good regex for a set of sequences. Especially for those who have little experience on formal expressions, a single mistake in regex composition leads to an invalid regex or, even worse, a regex with different semantics to ones' intention. An automatic regex synthesis tries to moderate this human error.

Examples (Oncina & García, 1992; Bartoli et al., 2014) and natural language descriptions (Locascio et al., 2016; Zhong et al., 2018b) are the two major sources of regex synthesis. Examples are specific in their meaning without any implicit semantics, but are hard to provide the complete user intent described by a regex. On the other hand, natural language descriptions are ambiguous but describes the high-level concept of a regex. As these two different types of information are complementary to each other, there have been many studies (Chen et al., 2020; Li et al., 2021; Zhong et al., 2018a) that suggest to use both modalities as they can use examples to fix the ambiguous expressions generated from natural language descriptions.

The *regex synthesis problem* is to make a proper regex $E$ from both positive and negative sets such that all the positive samples are generated by $E$ and all the negative strings are not generated by $E$. This problem is also called the regex *inference* or *learning* problem in the literature.

The traditional approaches for the regex synthesis problem are often inapplicable in practical settings. For example, Angluin's (1987) $L^*$ is impractical because it requires the equivalence test between a target regex and a generated regex, which is known to be PSPACE-complete and thus intractable (Stockmeyer & Meyer, 1973). Gold (1978) shows that the state characterization approach is NP-complete and thus intractable. Heuristic approaches based on state-merging DFA induction algorithms such as RPNI (Oncina & García, 1992) and blue-fringe (Lang et al., 1998) are not very promising on synthesizing complex regexes because it is almost impossible to obtain a concise regex even when they are able to produce a concise DFA from examples. As recent approaches, both AlphaRegex (Lee et al., 2016) and RegexGenerator++ (Bartoli et al., 2014; 2016) generate candidate regexes and match against the given examples until the target regex matches all examples successfully. While these approaches generate correct regexes when a time limit is very long, they often fail when the example size is large. In addition, both approaches do not use any characteristics in the given examples, which can be a useful feature to speed up the process. In other words, these methods are impractical for generating regexes within a reasonable time limit.

For practical usage, we notice that the divide-and-conquer approach is promising for synthesizing formal representation. For example, Alur et al. (2017) synthesize a program with hierarchical partition of the given input-output pairs. A decision tree reflects the hierarchical partition forms the control flow of the synthesized program. This method noticeably improves program synthesis time due to the partitioned problem space. Farzan & Nicolet (2021a;b) propose an effective method that converts a program into an equivalent program by splitting a problem into several subproblems. This divide-and-conquer approach is efficient since individual tasks for solving subproblems can run simultaneously. Recently, Barke et al. (2020); Shrivastava et al. (2021) propose program synthesis methods that first finds partial solutions satisfying subsets of input-output examples and combines the solutions to find the global solution.

In this paper, we propose a novel regex synthesis framework based on the divide-and-conquer paradigm. Given a set of positive examples, we split examples into multiple parts by grouping similar substrings using a neural network and solve subproblems defined over the multiple parts in serial order. We train a neural network called the *NeuralSplitter* that embeds a set of positive examples into a fixed-size vector and produces the most probable split for each example in the set.

Our main contributions are as follows:

1. We propose a regex synthesis framework *SplitRegex* based on the divide-and-conquer paradigm, which has been employed in many sequential algorithms. As far as we are aware, this is the first attempt to synthesize regexes in the divide-and-conquer manner.

2. We design a neural network called the *NeuralSplitter*—a two-stage attention-based recurrent neural network trained to predict the most probable split of input positive strings.

3. We conduct exhaustive experiments on various benchmark datasets including random dataset and practical regex datasets for demonstrating the effectiveness of the the proposed framework. We show that our framework can be easily combined with the off-the-shelf regex synthesis algorithms with minor modifications.

Namely, our framework solves regex synthesis problem, which is PSPACE-complete, much faster when applied to any existing regex synthesis models, and thus boosts the overall performance.

## 2 RELATED WORK

We provide a list of related works on regex synthesis problems based on various input types including positive and negative examples and natural language descriptions.

### 2.1 REGULAR EXPRESSION INFERENCE FROM EXAMPLES

One approach to regular expression inference is by Angluin's (1987). Angluin suggests an algorithm $L^*$ that infers an unknown DFA in teacher-student architecture. The teacher of $L^*$ offers

membership queries and equivalent queries on the target expression for the student. Note that this type of algorithms generally requires an infinite amount of examples to synthesize an FA.

Gold (1978) proposes a heuristic inference algorithm that inductively distinguishes random prefixes according to the acceptance followed by the given examples. The prefixes with the same acceptance form a single state in a DFA. The blue-fringe algorithm (Lang et al., 1998) and RPNI (Oncina & García, 1992) are similar to the Gold's algorithm.

AlphaRegex (Lee et al., 2016) exhaustively searches through all possible regular expressions until it identifies a solution regex that satisfies given positive and negative examples. A major limit of AlphaRegex is that its runtime is extremely due to the exponential increase of search space.

RegexGenerator++ (Bartoli et al., 2014; 2016) produces a regular expression using the genetic programming approach. The produced regular expression recognizes the surroundings (context) of the target sequence as well as the target itself. However, RegexGenerator++ entails a huge amount of computations due to the repetitive nature of genetic approach on each inference.

## 2.2 REGULAR EXPRESSION SYNTHESIS FROM NATURAL LANGUAGE DESCRIPTIONS

Ranta (1998) studies rule-based techniques for the conversion between multi-languages and regular expressions. Kushman & Barzilay (2013) build a parsing model that translates a natural language sentence to a regular expression, and provides a dataset, which is the most popular benchmark dataset in recent research. Locascio et al. (2016) propose the Deep-Regex model based on Seq2Seq for generating regular expressions from NL descriptions together with 10,000 NL-RX pair data. However, due to the limitations of the standard Seq2Seq model, the Deep-Regex model can only generate regular expressions similar in shape to the training data. The SemRegex model (Zhong et al., 2018b) improves the Deep-Regex model by reinforcement learning based on the DFA-equivalence oracle, which determines if two regular expressions define the same language, as a reward function and their model can generate correct regular expressions even if they do not exactly look like answers. The SoftRegex model (Park et al., 2019) further improves SemRegex based on the fast regex equivalence test, which gives rise to a fast learning process.

## 2.3 MULTI-MODAL REGULAR EXPRESSION SYNTHESIS

Most multi-modal synthesis method uses natural language descriptions and positive/negative examples simultaneously to mitigate the ambiguity in natural languages. Chen et al. (2020) introduce hierarchical sketch for the semantics of natural description. Chen et al.'s implementation Regel first deduces the hierarchical structure representing the semantics of the natural language description. This semantic structure has some holes which the regular expression inference algorithm enhances a regular expression by predefined rules. Regel also prunes an infeasible expression using over- and under-approximation possible regular expressions, which replaces the holes with the most permissive and the most restrictive expression, respectively. Li et al. (2021) propose TransRegex framework that synthesizes a valid regular expression. TransRegex's NLP-based synthesizer translates the given description to a regular expression and fixes the expression by rules and RFixer (Pan et al., 2019) to increase matching score on the examples.

## 2.4 REPAIR-BASE REGULAR EXPRESSION SYNTHESIS

Pan et al. (2019) design a heuristic algorithm RFixer that repairs a regular expression with given examples. RFixer improves the given regular expression by introducing or removing a hole and approximate the target expression with over- and under-approximation method until finds a regular expression with no holes that satisfies the given examples. Li et al. (2020) propose FlashRegex that generates a deterministic regular expression, which is a subclass of regular expressions. A SAT solver designs the basic FA structure after the given examples and FlashRegex iteratively shuffles the transitions in FA model at random to find the most suitable FA that satisfies the given examples.

## 3 OUR APPROACH

Our approach involves the following procedure:

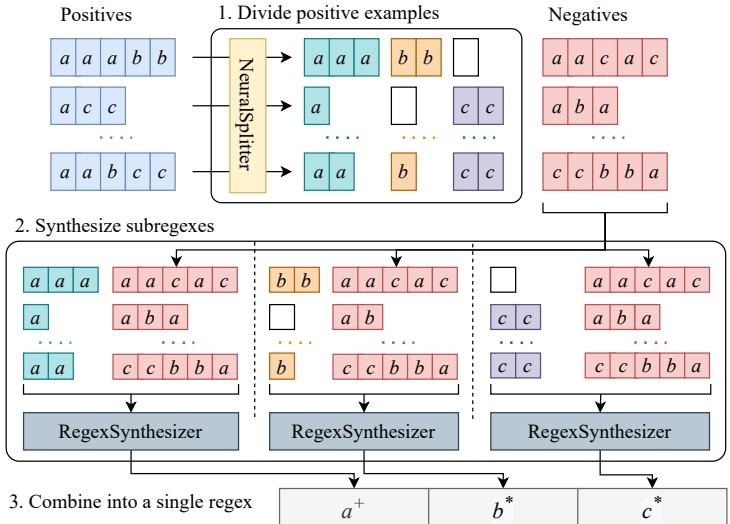

Figure 1: Overview of the proposed approach.

1. Split the input positive strings in the set $P$ into multiple partitions $P_1, P_2, \ldots, P_S$, where $S$ is the number of splits, using the NeuralSplitter model.

2. Run the regex synthesis engine for each $P_i$ for $1 \leq i \leq S$ with a corresponding set $N_i$ of negative strings. We will explain in more detail about how to use the set $N$ of negative strings for final regex synthesis.

3. Concatenate the obtained subregexes $R_1, R_2 \ldots, R_S$ such that $P_i \subseteq L(R_i)$ to produce the final regex $R$. Note that $L(R_i)$ is the language represented by $R_i$.

In order to train the NeuralSplitter, we generate target labels for positive examples as follows. For instance, when we are given $P = \{\text{aabbccabca}, \text{abcbbc}, \text{bbbcabb}, \text{aabbbc}\}$ and $R = \text{a*b*c*.*}$, we obtain ground truth labels for the four positive examples in $P$ as follows: $\{1122330000, 123000, 2223000, 112223\}$. The digits $1$, $2$, and $3$ in the ground truth labels imply that the corresponding symbols are generated from the first, second, and third subregex of $R$. Note that the output label for symbols generated from the wildcard pattern regex .* (capturing zero or more any characters) is always $0$ regardless of the position of the corresponding subregex. We explain the reason later in this section.

Note that we cannot split negative examples just as we do for positive examples using the NeuralSplitter model as there may be another split (not produced by the NeuralSplitter) that results in negative strings recognized by the synthesized regex. For instance, consider a simple case when given $P = \{\text{abbaaa}, \text{abaaa}, \text{aaa}\}$ and $N = \{\text{aaaa}\}$. The NeuralSplitter may simply predict that substrings with the same repeated symbols should be grouped to form a subregex. Then, the first part has positive examples $P_1 = \{\text{a}\}$ and $N_1 = \{\lambda\}$, the second part has $P_2 = \{\text{bb}, \text{b}, \lambda\}$ and $N_2 = \{\lambda\}$, and finally the last part has $P_3 = \{\text{aa}, \text{aaa}\}$ and $N_3 = \{\text{aaaa}\}$. It is possible that the regex synthesis algorithm generates $\text{ab}^*\text{aaa}^?$, which accepts the negative example.

We resolve this problem by introducing the concept of *prefix-conditioned subregex synthesis*. The basic idea of our approach is to first synthesize $n-1$ subregexes once the NeuralSplitter produces $n$ splits $(P_1, P_2, \ldots, P_n)$ from $P$. Then, we concatenate the resulting $n-1$ regexes to be $R_1 R_2 \cdots R_{n-1}$ and use this regex to obtain the final regex from the last split sample $P_n$ that satisfies all prefix subregexes as well as the negative samples. Namely, for each regex synthesis $R_i$, we only consider the corresponding split examples $P_i$ from $P$. We use the negative samples to prevent for a regex synthesis algorithm to return a trivial regex such as the wildcard pattern regex .* (capturing zero or more any characters). One may consider to use the NeuralSplitter approach for the negative samples. Our empirical observation confirms that it causes substantially larger amount of additional computations, and the performance becomes worsened.

When there are common strings between the split positive examples $P_i$ and the negative examples $N$, we remove the common strings from $N$. If $P_i$ has only one example (singleton set), then we simply returns the example as $R_i$. The proposed method generates a regex $R$ from a set $P$ of positive examples and a set $N$ of negative examples that satisfies the following conditions:

- $L(R_i) \subseteq P_i$ and $L(R_i) \cap (N \setminus P_i) = \emptyset$ for $1 \leq i \leq n-1$ (independent subregex synthesis).
- $L(R_1 R_2 \cdots R_n) \cap N = \emptyset$ (prefix-conditioned subregex synthesis).
- $L(R_i) = P_i$ if $|P_i| = 1$ for $1 \leq i \leq n$ (singleton subregex synthesis).

While the proposed method can generate a regex quickly by dividing a complex synthesis problem into several easier problems, one limit is that it cannot generate the wildcard pattern regex, .*, as a subregex $R_i$ due to the negative examples. Because the wildcard pattern is common and handy in practice to represent complicated sequences, we treat the symbols generated from the wildcard pattern differently compared with the rest of the symbols—we assign a special label 0 for the symbols in the data preprocessing step. For instance, if a string abbccdd is generated from a\*b\*c\*d\*, then the label is 1223344. However, if abbccdd is generated from a\*.\*, then the label is 1000000. When reading these special symbol 0's, our framework immediately generates wildcard patterns without going through the prefix-conditioned subregex synthesis.

## 3.1 DATA PREPROCESSING

From the practical regex dataset, we exclude regexes with features listed below:

- Backreference: we do not consider backreference for regex synthesis as some baseline approaches such as AlphaRegex and Blue-Fringe algorithm do not support regex features that capture non-regular languages.
- Lookahead/lookbehind: while regexes with these features still only capture regular languages, we do not consider these features as they require the regex engine to perform backtracking that causes catastrophic runtime in some extreme cases.
- Quantifier with exact count '{n}' or range '{n, m}': we replace numerical quantifiers with '\*' to 1) reduce the search space and 2) control the length of randomly generated strings.

For each regex, we randomly generate 20 positive and negative examples up to certain length (10 for random dataset and 15 for the other datasets). In order to generate positive examples, we utilize a Python library called the Xeger[1]. For generating negative examples, we first generate positive examples and randomly substitute symbols with other symbols to make the examples outside the language represented by the regex. Then, we use Python library pyre2, which is a Python wrapper for Google's RE2 regex library, to perform matchings of perturbed examples against the regex. We find this is effective compared to the complete random generation of negative examples.

In order to guarantee that every regular expression has 20 positive and negative examples, we exclude regular expressions that do not generate more than 20 positive strings (e.g., regexes without quantifiers such as \* and +). From 20 positive and negative strings, we use 10 positive/negative strings for generating regexes and use the remaining strings for measuring the semantic accuracy of synthesized regexes by membership queries.

We utilize the fullmatch method of pyre2 to find subregexes from the whole regex that match corresponding substrings from given positive strings.

## 3.2 NEURAL NETWORK ARCHITECTURE

We introduce a neural network called the *NeuralSplitter* that is trained to split positive examples generated from each regex. For a set $P = \{p_1, p_2, \ldots, p_n\}$ of positive examples, where $p_i \in \Sigma^*$ for $1 \leq i \leq n$, we embed each string $p_i$ with an RNN encoder into $d$ dimensional space as follows: $h_i = \mathrm{RNN}_{\mathsf{enc1}}(p_i)$, where $h_i \in \mathbb{R}^d$. Then, we embed the set $P$ of positive examples with the second RNN encoder $\mathrm{RNN}_{\mathsf{enc2}}$ as follows: $h_P = \mathrm{RNN}_{\mathsf{enc2}}(h_1, h_2, \ldots, h_n)$, where $h_P \in \mathbb{R}^d$.

---

[1] https://pypi.org/project/xeger/

From the hidden vectors $h_i$ for $1 \leq i \leq n$ and $h_P$, the RNN decoder is trained to produce the ground truth label for each positive string $p_i$ as follows:

$$o_1, o_2, \ldots, o_{l_i} = \text{RNN}_{\text{dec}}(W_{\text{dec}}^{\top} \cdot [h_i; h_P]),$$

where $l_i = |p_i|$, $o_j \in \mathbb{R}^d$ for $1 \leq j \leq l_i$, $W_{\text{dec}} \in \mathbb{R}^{2d \times d}$.

Finally, we retrieve the predicted label for each positive example $p_i$ as follows: $y_j = \text{softmax}(W_{\text{output}}^{\top} \cdot o_j)$, where $y_j \in \mathbb{R}^{|V|}$, $W_{\text{output}} \in \mathbb{R}^{d \times |V|}$, and $V$ is the output vocab. Note that $y_j$ is the predicted probability distribution for the label of $j$th symbol of the positive example $p_i$.

During the decoding process, the RNN decoder exploits the attention mechanism to better point out the relevant parts from the entire set of positive strings. Since the split label for a positive string relies on the information from all positive strings, we implement two-step attention mechanism that enables our model to attend both in set level and example level. For instance, when we have two strings aabbbc and aabbcc, the most probable labels for these two strings are 112223 and 112233. However, if the second string is caabbcc, where only the first symbol c is appended in front of aabbcc, then the most probable labels for two strings become 223334 and 1223344. In this reason we need to use the information of all given positive strings to correctly predict the output labels.

We use bi-directional GRU network for RNN encoder and decoder of NeuralSplitter. Cross entropy loss is used for training neural network.

## 4 Experimental Setup

We describe the experimental setup for evaluation.

**Datasets.** We use the following benchmarks to measure the performance of the proposed approach compared to the previous approaches.

- Random dataset: consists of 1,000,000 randomly generated regexes (784,682 unique regexes) over various alphabet sizes 2, 4, 6, 8 and 10.
- RegExLib[2]: consists of regexes collected from an online regex library where 4,149 regexes are indexed from 2,818 contributors to help regex users write their own regexes more easily.
- Snort[3]: an open source intrusion detection/prevention system, contains more than 1,200 regexes in its rule set for compactly describing malicious network traffic.
- Polyglot Regex Corpus (Davis et al., 2019): consists of 537,806 regexes extracted from 193,574 software projects written in eight popular programming languages. Introduced for the research on regex portability problem.

For training the NeuralSplitter and evaluating our framework on practical regex benchmarks including RegExLib, Snort and Polyglot Regex Corpus, we first generate 20 random positive and negative examples for each regex and split the data into training, validation and test set with the ratio of 8:1:1. Then, we collect the training and validation splits for three benchmarks for training the NeuralSplitter model, and use each test set for evaluating the regex synthesis performance on each benchmark.

**Hyperparameters.** We use two-layer bidirectional GRU with 256 hidden units unless explicitly mentioned otherwise. The dimension of input token embedding is four. The Adam optimizer is used with batch size 512 and learning rate of 0.001 and weight decay of 1e-6.

**Baselines.** We take the following baseline methods for regex synthesis from examples.

- AlphaRegex (Lee et al., 2016; Kim et al., 2021): a best-first search based regex enumeration algorithm. We use our own Python implementation of AlphaRegex algorithm, which is originally written in OCaml[4] together with the recent search space reduction techniques by Kim et al. (2021).

---

[2] https://www.regexlib.com/
[3] https://www.snort.org/
[4] https://github.com/kupl/AlphaRegexPublic

- RegexGenerator++ (Bartoli et al., 2014; 2016): a regex inference algorithm based on genetic programming. Due to the nature of genetic inference procedure, this is the slowest among the baselines.
- Blue-Fringe algorithm (Lang et al., 1998): a classical state-merging DFA induction algorithm. We choose this algorithm to show the generality of our framework on classical synthesizers such as $L^*$ (Angluin, 1987) or RPNI (Oncina & García, 1992).

Note that we also use these baselines as off-the-shelf regex synthesizers for our SplitRegex framework to evaluate the performance boost gained by our framework.

**Metrics.** We evaluate the performance of the proposed approach in two main aspects: 1) the time efficiency of synthesis algorithm and 2) the accuracy of synthesized regexes. For evaluating the time efficiency, we set a time budget, specified by a *timeout* value. We set the timeout value to 3 (seconds) for AlphaRegex and Blue-Fringe algorithm and 10 (seconds) for RegexGenerator++ in our experiments. A regex synthesis is counted as a *success* if an algorithm finds a regex satisfying given positive and negative examples within the specified timeout. We measure the *success rate* of regex synthesis by counting the number of successes in $n$ regex synthesis trials where $n$ is the number of target regexes in test set.

Meanwhile, we also need to measure the accuracy of synthesized regexes as any algorithm may generate a trivial solution that only recognizes positive examples. However, we cannot simply count the number of cases where the synthesized regex is 'literally' equivalent to the target regex as two different regexes can capture the same set of strings as there are infinitely many semantically equivalent regexes. Hence, we need to introduce a criterion that considers how two regexes are semantically similar. While it is well-known that it is possible to algorithmically decide whether two regexes represent the same language, it is also well-known that the decision process is PSPACE-complete (Stockmeyer & Meyer, 1973) and therefore probably intractable.

Therefore, we define a metric called the *semantic regex accuracy* that utilizes randomly generated positive and negative examples that are not used for regex synthesis. Remark that the semantic regex accuracy has been already employed in a previous work for evaluating the generated regexes (Li et al., 2021). Given a pair $(P, N)$ of positive and negative sets reserved for evaluation, we define the semantic regex accuracy as follows:

$$\text{sem\_accuracy}(R, P, N) = \frac{|T_P| + |T_N| - |F_P| - |F_N|}{|P| + |N|} \times 100,$$

where $T_P = \{w \in L(R) \mid w \in P\}, T_N = \{w \notin L(R) \mid w \in N\}, F_P = \{w \notin L(R) \mid w \in P\}$ and $F_N = \{w \in L(R) \mid w \in N\}$.

Intuitively, the semantic regex accuracy is higher if the synthesized regex recognizes more strings recognizable by the target regex and rejects more strings not recognizable by the target regex.

Finally, we define a synthesized regex to be *fully accurate* if the regex satisfies all the unseen examples. In other words, a regex is fully accurate if the semantic regex accuracy is 1.

## 5   EXPERIMENTAL RESULTS AND ANALYSIS

We demonstrate the experimental results to show the effectiveness of the proposed approach compared to baselines and analyze the results.

### 5.1   MAIN RESULTS

Table 1 shows the main result of our paper. We evaluate our approach with the baselines in terms of success rate, semantic accuracy, and the ratio of fully accurate regexes using reserved examples.

In order to examine the upper bound on the performance of SplitRegex, we also consider a variant of SplitRegex called 'GT Split' where we use ground-truth splits of regexes instead of predicted splits by NeuralSplitter.

It is readily seen that the proposed framework substantially improves all of the success rate, semantic accuracy, ratio of fully accurate regexes in most benchmarks except some cases.

Table 1: Regex synthesis performance (success rate, semantic regex accuracy, and the ratio of fully accurate regexes) of our model and baselines over three benchmark regex datasets. 'AR', 'BF', and 'RG' are abbreviations for AlphaRegex, Blue-Fringe, and RegexGenerator++, respectively.

| Method | RegExLib | | | Snort | | | Polyglot Corpus | | |
|---|---|---|---|---|---|---|---|---|---|
| | Succ. | Acc. | Full. | Succ. | Acc. | Full. | Succ. | Acc. | Full. |
| AlphaRegex | 14.0 | 13.56 | 12.1 | 16.2 | 15.74 | 13.8 | 49.9 | 49.04 | 46.2 |
| SplitRegex (+AR) | 23.4 | 21.78 | 17.6 | 72.1 | 67.42 | 54.4 | 64.0 | 62.28 | 56.4 |
| GT Split (+AR) | 43.8 | 40.90 | 32.3 | 87.7 | 83.20 | 69.6 | 70.4 | 67.68 | 59.6 |
| RegexGenerator++ | 0.0 | 0.00 | 0.0 | 0.0 | 0.00 | 0.0 | 0.2 | 0.20 | 0.2 |
| SplitRegex (+RG) | 2.4 | 2.34 | 2.1 | 6.4 | 5.68 | 3.4 | 17.0 | 16.82 | 16.0 |
| GT Split (+RG) | 2.9 | 2.80 | 2.2 | 9.3 | 8.56 | 5.8 | 17.6 | 17.26 | 16.0 |
| Blue-Fringe | 100.0 | 0.80 | 0.1 | 100.0 | 0.46 | 0.0 | 100.0 | 4.40 | 0.3 |
| SplitRegex (+BF) | 86.9 | 15.26 | 1.3 | 97.3 | 17.22 | 2.6 | 97.4 | 13.32 | 1.7 |
| GT Split (+BF) | 90.8 | 18.00 | 2.1 | 100.0 | 19.42 | 3.0 | 99.4 | 13.72 | 1.7 |

Table 2: Comparisons of different regex synthesis method using split positive examples.

| Synthesis Strategy | On Random Regexes | | On Practical Regexes | |
|---|---|---|---|---|
| | Success Rate | Accuracy | Success Rate | Accuracy |
| Baseline (AlphaRegex) | 47.57 | 44.83 | 12.90 | 13.08 |
| Prefix-conditioned synthesis | 63.90 | 60.50 | 37.60 | 36.94 |
| Independent parallel synthesis | 42.00 | 40.40 | 54.50 | 54.34 |
| Independent sequential synthesis | **64.60** | **61.68** | **55.84** | **55.20** |

First, the success rate of the Blue-Fringe algorithm is the highest among all the considered algorithms due to the nature of the Blue-Fringe algorithm. Note that the Blue-Fringe algorithm merges the suffixes of given positive examples while assuming that any string the expression represents is positive unless there is a counterexample—a negative example. While the Blue-Fringe algorithm can simply generate a regex satisfying the examples very quickly, the generated regex represents a slightly larger set of strings than the set of positive examples in general. This means that the regex generated by Blue-Fringe rarely accepts positive examples that are not used in the regex synthesis process. However, when the SplitRegex framework is combined with Blue-Fringe, the semantic accuracy is improved even though it fails to return solutions for some cases due to additional overhead caused by neural network computations. The main reason is that the process of splitting examples already makes the resulting regexes more fine-grained compared to those generated by the vanilla Blue-Fringe.

The performance on RegexGenerator++ is very low across all experiments mostly due to the repetitive nature of genetic approach. Actually, RegexGenerator++ fails to generate regexes in most cases even with larger time limit (10 seconds). Nevertheless, we can see that the proposed framework helps RegexGenerator++ succeed in a few more cases compared to the vanilla RegexGenerator++.

## 5.2 DETAILED ANALYSIS

**Alphabet size.** Figure 2 shows how the SplitRegex framework affects the regex synthesis performance of AlphaRegex and Blue-Fringe algorithms on random dataset with different alphabet sizes. While the success rate decreases as the alphabet size increases for AlphaRegex, SplitRegex effectively improves the success rate as the alphabet size increases. Moreover, the semantic accuracy of synthesized regexes is also improved when the SplitRegex is applied to both baselines. This is because it is easier to group the similar substrings using common symbols as there are more symbols in a string. For example, two regexes $abc * d?(e + f)$ and $000 * 1?(0 + 1)$ are similar in terms of length and structure but it is easier to find correct split from the examples generated from the former as we can infer where each symbol in the examples comes from more easily.

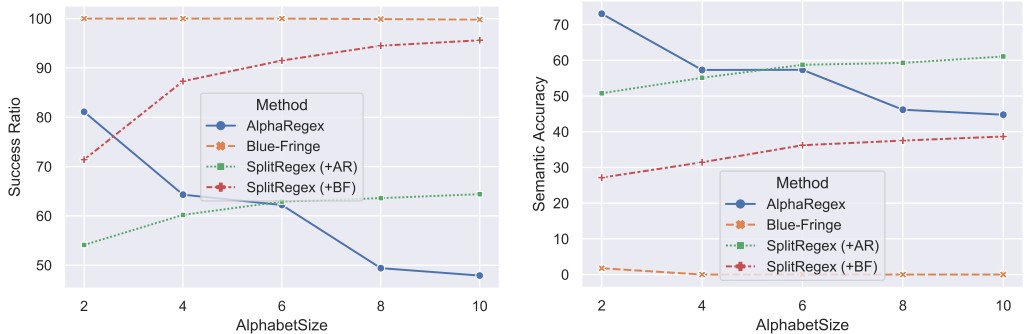

Figure 2: Results on random regular expression dataset with different alphabet sizes.

Table 3: Examples of neural example splitting.

| Dataset | Input String | Split Result | | | | |
|---------|--------------|------|---|---|---|---|
| RegExLib | uname=&pword=9E8 | uname= | | &pword= | 9E8 | |
| | uname=!&pword= | uname= | ! | &pword= | | |
| | uname=Hj%}v&pword='Af7= | uname= | Hj%}v | &pword= | 'Af7= | |
| | uname=)#&pword=@k | uname= | )# | &pword= | @k | |
| Snort | /images!php?t=6153/U | /images | ! | php?t= | 6153 | /U |
| | /images-php?t=296/U | /images | - | php?t= | 296 | /U |
| | /images4php?t=5213/U | /images | 4 | php?t= | 5213 | /U |
| | /imagesQphp?t=165/U | /images | Q | php?t= | 165 | /U |

**Subregex synthesis strategies.** We conduct additional experiments to verify whether the 'prefix-conditioned synthesis' strategy can be utilized for every subsequent subregexes while we utilize only for synthesizing the last subregex. Moreover, we also test whether the first $n-1$ subregexes (when the split size is $n$) can be synthesized in parallel if the prefix-conditioned synthesis is not used. Table 2 demonstrates the experimental results on random regexes and practical regexes from RexExLib, Snort and Polyglot corpus datasets. The results imply that the simplest approach (independent sequential synthesis except the final subregex) achieves the best performance compared to the parallel approach and prefix-conditioned synthesis approach.

**Examples of neural splitting.** Table 3 demonstrates several examples of neural example splitting. For instance, positive examples in the 'Snort' dataset is generated from a regex '/images.*php?t=.*/U'. We can see that our NeuralSplitter model successfully splits positive examples into five parts where similar strings are grouped together. As a result, SplitRegex produces the following regex for the examples which is quite similar to the target regex: '/images.*php?t=[0-9]*/U'.

## 6 CONCLUSIONS AND FUTURE WORK

In this paper, we have proposed a regex synthesis framework called the SplitRegex that synthesizes a regex from given positive and negative examples based on the divide-and-conquer paradigm. Given a set of positive examples, we predict the most probable splits for strings such that substrings in the same partition exhibit similar patterns using a neural network called the NeuralSplitter. Then, we synthesize subregexes for all partitions and derive the final regex satisfying the input examples. Our experimental results show that the proposed scheme is very effective in reducing the time complexity of regex synthesis and actually yields more accurate regexes than the previous approaches.

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

APPENDIX

## A BACKGROUND

Here we provide some background knowledge on the task of regular expression synthesis/inference and basic neural network architectures used in our model.

**Regular expression inference/synthesis.**    Let $\Sigma$ be a finite alphabet and $\Sigma^*$ be the set of all strings over the alphabet $\Sigma$. A *regular expression* (regex) over $\Sigma$ is $a \in \Sigma$, or is obtained by applying the following rules finitely many times. For regexes $R_1$ and $R_2$, the union $R_1 + R_2$, the concatenation $R_1 \cdot R_2$, the star $R_1^*$, and the question $R_1^?$ are also regexes. Note that $L(R_1^?)$ is defined as $L(R_1) \cup \{\epsilon\}$.

Given a set of positive and negative strings, we consider the problem of synthesizing a concise regex that is consistent with the given strings. The examples are given by a pair $(P, N)$ of two sets of strings, where $P \subseteq \Sigma^*$ is a set of *positive strings* and $N \subseteq \Sigma^*$ is a set of *negative strings*.

Then, our goal is to find a regex $R$ that accepts all positive strings in $P$ while rejecting all negative strings in $N$. Formally, $R$ satisfies the following condition: $P \subseteq L(R)$ and $L(R) \cap N = \emptyset$.

**Recurrent neural networks.**    An RNN is described by a function that takes an input $x_t$ and a hidden state $h_{t-1}$ at time step $t - 1$ and returns a hidden state $h_t$ at time step $t$ as follows: $h_t = \mathrm{RNN}(x_t, h_{t-1})$.

Given an input sentence $\mathbf{x} = (x_1, x_2, \dots, x_n)$, an encoder $\mathrm{RNN_{enc}}$ plays its role as follows: $h_t = \mathrm{RNN_{enc}}(x_t, h_{t-1})$ where $h_t \in \mathbb{R}^d$ is a hidden state at time $t$. After processing the whole input sentence, the encoder generates a fixed-size context vector that represents the sequence as follows: $c = q(h_1, h_2, \dots, h_n)$.

Usually, $q$ simply returns the last hidden state $h_n$ in one of the original sequence to sequence paper by Sutskever et al. (2014). Since the initial hidden state $h_0$ is usually randomly initialized, we simply define RNN as a function that takes sequence $\mathbf{x}$ and produces a vector $c$ as follows: $c = \mathrm{RNN}(\mathbf{x})$.

Now suppose that $\mathbf{y} = (y_1, y_2, \dots, y_m)$ is an output sentence that corresponds to the input sentence $\mathbf{x}$ in training set. Then, the decoder RNN is trained to predict the next word conditioned on all the previously predicted words and the context vector from the encoder RNN. In other words, the decoder computes a probability of the translation $\mathbf{y}$ by decomposing the joint probability into the ordered conditional probabilities as follows:

$$p(\mathbf{y}) = \prod_{i=1}^{m} p(y_i | \{y_1, y_2, \dots, y_{i-1}\}, c).$$

Now our decoder $\mathrm{RNN_{dec}}$ computes each conditional probability as follows:

$$p(y_i | y_1, y_2, \dots, y_{i-1}, c) = \mathrm{softmax}(W^\top \cdot s_i),$$

where $s_i \in \mathbb{R}^d$ is the hidden state of decoder RNN at time $i$ and $W \in \mathbb{R}^{d \times |V|}$ is a linear transformation that outputs a vocabulary-sized vector. Note that the hidden state $s_i$ is computed by

$$s_i = \mathrm{RNN_{dec}}(y_{i-1}, s_{i-1}, c),$$

where $y_{i-1}$ is the previously predicted word, $s_{i-1}$ is the last hidden state of decoder RNN, and $c$ is the context vector computed from encoder RNN.

**Attention mechanism.**    In order to resolve the long-range dependency problem in the sequence-to-sequence model, Bahdanau et al. (2015) defined each conditional probability at time $i$ depending on a dynamically computed context vector $c_i$ as follows:

$$p(y_i | y_1, y_2, \dots, y_{i-1}, \mathbf{x}) = \mathrm{softmax}(g(s_i)),$$

where $s_i$ is the hidden state of the decoder RNN at time $i$ computed by $s_i = \mathrm{RNN_{dec}}(y_{i-1}, s_{i-1}, c_i)$.

The context vector $c_i$ is computed as a weighted sum of the hidden states from encoder: $c_i = \sum_{j=1}^{n} \alpha_{ij} s_j$, where

$$\alpha_{ij} = \frac{\exp(\mathrm{score}(s_i, h_j))}{\sum_{k=1}^{n} \exp(\mathrm{score}(s_k, h_j))}.$$

Encoder: encode positive examples

Decoder: predict split labels for input positive examples

Figure 3: Neural network architecture of the 'NeuralSplitter' model.

Table 4: Statistics of four benchmark datasets. Numbers in parentheses are the numbers of regexes in original datasets and in front of the parentheses are the numbers of regexes used in our experiments.

| Dataset | # of Regexes | Alphabet Size | Max Length |
|---|---|---|---|
| Random | 784,682 | $\{2, 4, 6, 8, 10\}$ | 81 |
| RegExLib | 1,422 (3,072) | 128 | 767 |
| Snort | 375 (1,254) | 128 | 203 |
| Polyglot Corpus | 199,350 (537,800) | 128 | 1,903 |

Here the function 'score' is called an *alignment function* that computes how well the two hidden states from the encoder and the decoder, respectively, match. For example, $\text{score}(s_i, h_j)$, where $s_i$ is the hidden state of the encoder at time $i$ and $h_j$ is the hidden state of the decoder at time $j$ implies the probability of aligning the part of the input sentence around position $i$ and the part of the output sentence around position $j$.

## B  NEURAL NETWORK ARCHITECTURE

Figure 3 depicts the neural network architecture of the 'NeuralSplitter' model. Note that both RNN encoders used in our neural network are bi-directional.

## C  EXPERIMENTAL DETAILS

### C.1  STATISTICS OF DATASETS

We refer the readers to Table 4 for the statistics of benchmark datasets. For random dataset, we randomly generate in total 1,000,000 regexes independently where there are 200,000 regexes for each alphabet size in $\{2, 4, 6, 8, 10\}$. The number of unique regexes is 784,682 as we allow duplicates in random generation.

Table 5: Split performance (%) of the NeuralSplitter under different settings of hyperparameters.

| Dataset | RNN Cell | Dimension | # of Layers | String Accuracy | Set Accuracy |
|---------|----------|-----------|-------------|-----------------|--------------|
| Random | GRU | 64 | 1 | 54.10 | 35.04 |
| | | | 2 | 55.34 | 37.04 |
| | | 128 | 1 | 57.41 | 38.03 |
| | | | 2 | 57.41 | 38.03 |
| | | 256 | 2 | 57.41 | 38.24 |
| | | 384 | 2 | 57.21 | 38.25 |
| | LSTM | 256 | 2 | **57.93** | **38.34** |
| Practical | GRU | 256 | 2 | **76.78** | **68.20** |

**Environment.**    We conduct experiments on a server equipped with Intel Core i7-8700K processor with 6 cores, 48GB DDR4 memroy, and NVIDIA GTX 1080 Ti GPU. We use the neural network framework PyTorch for experiments.

# D    ADDITIONAL RESULTS

## D.1    PERFORMANCE OF 'NEURAL EXAMPLE SPLITTING'

The performance of the NeuralSplitter model under different settings of hyperparameters and RNN cell types is provided in Table 5. We define two metrics, 'string accuracy' and 'set accuracy', for evaluating the performance of neural example splitting. String accuracy is the ratio of correctly labeled examples from ten positive examples for each test sample. Set accuracy means the ratio of correctly labeled sets as a whole. If the NeuralSplitter correctly splits ten positive examples for a test sample, then it is regarded as 'correctly labeled'. It is notable that the NeuralSplitter performs better on practical dataset which is apparently more complex than random dataset.

While we find that LSTM cell performs slightly better than GRU cell when the other hyperparameters are equivalent, we decide to use GRU instead of LSTM as GRU is known to execute faster than LSTM. We also find that the performance of the NeuralSplitter is higher as we increase the number of hidden units and the number of layers until 256 and 2, respectively. We decide not to increase the number of parameters further as the computation efficiency is also a very important metric in the task of regex synthesis.

## D.2    PERFORMANCE COMPARISON WITH DIFFERENT TIMEOUT VALUES

Table 6 shows how the regex synthesis performance changes as the timeout value changes. We perform regex synthesis with various timeout values 1, 2, 3, 5, and 10 seconds and measure the success ratio and semantic accuracy of the proposed baselines including our approach. We do not include RegexGenerator++ for comparison as it fails to synthesize regexes in most cases even with timeout value of 10 seconds. The result shows that Blue-Fringe algorithm succeeds in regex synthesis within 1 second in most cases as the success rate and semantic accuracy does not increase with higher timeout values.

On the other hand, AlphaRegex with and without our SplitRegex framework exhibits better regex synthesis success rate and semantic accuracy as timeout value increases.

## D.3    SPLIT VS NON-SPLIT

We evaluate the performance of our SplitRegex framework by comparing the results of regex synthesis model with and without the SplitRegex framework for each test case. First, we present the number of test cases that are exclusively solved by each method. 'Win Ratio' means the ratio of test cases where each method performs better (synthesizes regex faster). We do not count the cases where both methods fail to synthesize regexes when calculating 'Win Ratio (%)'.

Table 6: Regex synthesis performance (success rate and semantic regex accuracy) of our model and baselines over four benchmark regex datasets under different timeout values. 'AR' is abbreviations for AlphaRegex. 'Random' dataset consists of regular expressions over alphabet of size ten.

| Method | Time. | Random | | RegExLib | | Snort | | Polyglot | |
|---|---|---|---|---|---|---|---|---|---|
| | | Succ. | Acc. | Succ. | Acc. | Succ. | Acc. | Succ. | Acc. |
| AlphaRegex | 1 | 34.67 | 32.03 | 9.67 | 9.63 | 8.33 | 8.01 | 42.67 | 42.29 |
| SplitRegex (+AR) | | 58.33 | 55.21 | 20.00 | 19.24 | 71.67 | 67.47 | 64.00 | 62.34 |
| AlphaRegex | 2 | 41.00 | 38.20 | 12.33 | 12.11 | 9.00 | 8.46 | 46.33 | 45.91 |
| SplitRegex (+AR) | | 66.00 | 62.20 | 20.67 | 19.63 | 72.67 | 68.27 | 64.67 | 62.97 |
| AlphaRegex | 3 | 45.67 | 42.47 | 13.00 | 12.76 | 13.67 | 12.79 | 51.00 | 50.16 |
| SplitRegex (+AR) | | 67.67 | 63.79 | 21.33 | 19.83 | 73.00 | 68.56 | 65.67 | 63.57 |
| AlphaRegex | 5 | 51.00 | 47.14 | 14.00 | 13.76 | 14.33 | 13.43 | 53.67 | 52.69 |
| SplitRegex (+AR) | | 72.67 | 68.47 | 21.67 | 20.09 | 73.67 | 69.07 | 66.33 | 63.91 |
| AlphaRegex | 10 | 55.00 | 50.94 | 19.00 | 18.56 | 16.00 | 14.84 | 57.33 | 56.13 |
| SplitRegex (+AR) | | 74.00 | 69.54 | 22.00 | 21.22 | 74.33 | 69.51 | 66.33 | 63.91 |

Table 7: Comparisons of regex synthesis performance of AlphaRegex algorithm with and without SplitRegex framework on practical regex datasets.

| Method | Success Ratio | Win Ratio | Runtime | Runtime for Success Cases |
|---|---|---|---|---|
| AlphaRegex | 25.77 | 20.82 | 2.4344 | 0.1250 |
| SplitRegex (+AR) | **46.10** | **79.18** | **1.6622** | **0.0088** |

We also estimate the average amount of time taken for regex synthesis in two different cases. First, 'Runtime (seconds)' is the average amount of time taken for synthesizing regexes in all test cases. If the method fails to synthesize a regex, then we consider it as '3 seconds' which is the timeout value used for this experiment. Second, 'Runtime for Success Cases (seconds)' is the average amount of time taken for synthesizing regexes in cases where both methods succeeds in synthesizing regexes.

Experimental results on practical regex datasets (RegExLib, Snort and Polyglot Corpus dataset) are provided in Table 7. Results show that SplitRegex sufficiently improves the success ratio and reduces runtime complexity taken for regex synthesis in almost 80% of test cases.

### D.4 USING NEGATIVE EXAMPLES GENERATED FROM RANDOMLY PERTURBED REGEXES

As we mentioned in the second paragraph of Section 3.1, we generate negative examples by randomly substituting symbols from positive examples. We call this method 'Symbol Perturb' here.

In order to generate 'harder' negative examples, we introduce a noise at the level of regex and call the method 'Regex Perturb'. We perform the 'Regex Perturb' by randomly choosing one of the following three manners: 1) Randomly substitute a symbol in the target regex, 2) Randomly insert a subregex from the target regex, and 3) Randomly delete a subregex from the target regex.

Table 8 shows the results. We observe that our method shows better performance than the baseline (AlphaRegex) for both cases. It is also seen that the 'Regex Perturb' generates harder negative examples than 'Symbol Perturb' as the performance is lower for negative examples generated by 'Regex Perturb' in general.

### D.5 PERFORMANCE ON DATASETS FROM TRANSREGEX (LI ET AL., 2021)

We conduct additional experiments on datasets—KB13 and NL-RX-Turk—used in TransRegex Li et al. (2021), a previous work on the multi-modal regex synthesis from both natural language

Table 8: Comparisons of regex synthesis performance evaluated for negative examples generated from randomly perturbed regexes.

| Method | Symbol Perturb | | | Regex Perturb | | |
|---|---|---|---|---|---|---|
| | Succ. | Acc. | Full. | Succ. | Acc. | Full. |
| AlphaRegex | 46.20 | 42.60 | 28.40 | 24.00 | 22.70 | 17.50 |
| SplitRegex (+AR) | 63.20 | 59.72 | 45.30 | 38.75 | 36.25 | 29.00 |
| GT Split (+AR) | 76.40 | 73.78 | 61.90 | 57.75 | 54.53 | 44.75 |

Table 9: Regex synthesis performance of our model in datasets used in TransRegex (Li et al., 2021).

| Method | KB13 | | | NL-RX-Turk | | |
|---|---|---|---|---|---|---|
| | Succ. | Acc. | Full. | Succ. | Acc. | Full. |
| AlphaRegex | 77.33 | 77.33 | 77.33 | 51.33 | 50.53 | 48.33 |
| SplitRegex (+AR) | 67.00 | 66.94 | 66.66 | 55.66 | 55.54 | 55.00 |
| GT Split (+AR) | 80.00 | 80.00 | 80.00 | 72.00 | 71.80 | 71.00 |

Table 10: Additional examples of neural example splitting.

| Dataset | Input String | Split Result | |
|---|---|---|---|
| | Invalid!v%alias@Z | Invalid | !v%alias@Z |
| | InvalidaccountD#f | Invalid | accountD#f |
| Polyglot | Invalid{pv_address | Invalid | {pv_address |
| | InvalidvSemail5 C | Invalid | vSemail5 C |
| | InvalidaH2'mailboxvb4A | Invalid | aH2'mailboxvb4A |

descriptions and examples. We found that regexes used in our experiments are more complex and use a larger set of symbols. The only difference is that some regexes from KB13 and NL-RX-Turk have 'and' and 'not' operators that are not supported by the standard regex library (pyre2) used in our experiments. For this reason, we conduct experiments on regexes from KB13 and NL-RX-Turk without 'and' and 'not' operators. Table 9 shows the experimental results. We can see that the performance of the proposed method is higher for NL-RX-Turx dataset but lower for KB13 dataset. We speculate that the main reason of the poor performance on KB13 is the size of KB13 dataset. Note that there are only 814 regexes in KB13 and we use only 440 regexes among the 814 regexes as the rest of regexes have 'and' or 'not' that are not compatible with standard regex library.

## D.6 ADDITIONAL EXAMPLES OF NEURAL EXAMPLE SPLITTING

In Table 10, we present additional examples of neural example splitting to exhibit the upper limit on the hardness of regexes the proposed method cannot generate.

Note that here the target regex is 'Invalid.*(account|address|email|mailbox).*. Since we rely on at most 10 positive examples to perform neural example splitting, it is difficult to find a split corresponding to the 'union' operator with many subregexes.

