# OpenReview forum: "SplitRegex: Faster Regex Synthesis via Neural Example Splitting"
_ICLR.cc/2022/Conference — ICLR 2022 Submitted_

### Official Review · Reviewer_fKY9 · 2021-11-02

**Correctness:** 3
**Technical Novelty And Significance:** 3
**Empirical Novelty And Significance:** 2
**Recommendation:** 5
**Confidence:** 4

**Main Review:**

Overall, I think that the proposed idea is itself simple, which means it should be easy to use, and at the same time it could potentially be useful for the task. However, I have several comments/questions, which limit my understanding and make the potential impact less clear. Resolving these would greatly strengthen the paper.

1. It is not clear if the proposed approach has an upper limit on the hardness regexes it can generate. Without this understanding, the usefulness of the approach is not clear.

2. Related to the previous question, how does the complexity of the regexes in this submission compare to the regexes used in the multi-modal settings such as Ye et al. (2020) and TransRegex (Li et al., 2021)? It is possible to evaluate the proposed method on their datasets while ignoring the textual inputs?

Benchmarking Multimodal Regex Synthesis with Complex Structures.
Xi Ye, Qiaochu Chen, Isil Dillig, and Greg Durrett. Proceedings of ACL 2020.

3. In the proposed approach, the results NeuralSplitter are final in the sense that after computing the output of the network (splits of the input examples) it cannot be changed and there is no search over these splits. It looks like an error of this stage can be fatal for the overall method. Evaluating how bad this effect is would strengthen the paper.

4. I don't understand why the results of the Blue-Fringe baseline are evaluated so differently w.r.t. the two metrics. Such difference should be easy to see qualitatively on several examples.

5. The overall performance of the SplitRegex (from table 1) is around 60%, which is higher than the baselines, but it still means that there are lots of errors. Showing and analyzing the error cases (on top of Successful examples in Table 3) would greatly strengthen the paper.

6. I also feel like another metric is missing: the percentage of the regexes that are generated correctly in the sense that all the unseen by the generation process positive and negative examples (the ones used in semantic regex accuracy) are recognized correctly within the time limit. This metric is stricter and will indicate how many regexes are actually correctly generated.

7. It looks like none of the baselines uses a neural network. Is it actually true that no published work put positive/negative regex examples into a neural network?

8. [Clarity] I did not find a description of how numerical targets of NeuralSplitter (like 1223344) were defined. I have my guesses but am not sure.

**Summary Of The Paper:**

This paper studies the problem of synthesizing regular expressions given positive/negative examples (no textual description). The core of the proposed method is a neural network that takes the examples and splits all of them into several parts. Then, an external search-based system is used to generate regex for each part. Finally, the regexes for the parts are concatenated into one overall regex.
The proposed system is evaluated against several baselines (the same external systems applied not to parts but to complete regexes) and is shown to bring benefits.

**Summary Of The Review:**

Overall, I think that the proposed idea is itself simple, which means it should be easy to use, and at the same time it could potentially be useful for the task. However, I have several comments/questions, which limit my understanding and make the potential impact less clear. Resolving these would greatly strengthen the paper. The issues lie at the intersection of clarity, positioning, and evaluation.

---

> ### Author Response · Authors · 2021-11-21
> **Response to reviewer fKY9. Part 1**
>
> We thank the reviewer for the valuable comments and detailed suggestions. We respond to the reviewer’s comments as follows.
>
> (1) It is not clear if the proposed approach has an upper limit on the hardness regexes it can generate. Without this understanding, the usefulness of the approach is not clear.
>
> - The regex datasets used in our experiments (especially polyglot corpus) actually contain regexes collected from real-world projects. Therefore we claim that the upper limit on the hardness of regexes can be defined by the set of excluded regexes in our experiments due to the existence of backreference, lookahead/lookbehind, and numerical quantifiers. Note that we excluded these regexes because all of the baseline algorithms such as AlphaRegex, Blue-Fringe, and RegexGenerator++ are not able to synthesize regexes with the features.
>
> (2) Related to the previous question, how does the complexity of the regexes in this submission compare to the regexes used in the multi-modal settings such as Ye et al. (2020) and TransRegex (Li et al., 2021)? It is possible to evaluate the proposed method on their datasets while ignoring the textual inputs?
>
> - We have checked the regexes mentioned in the comment and found that regexes used in our experiments are more complex and use a larger set of symbols than the regexes used in Ye et al. (2020) and TransRegex (Li et al. 2021). The only difference is that some regexes from KB13 and NL-RX-Turk have ‘and’ and ‘not’ operators that are not supported by the standard regex library (pyre2) used in our experiments. For this reason, we have conducted experiments on regexes from KB13 and NL-RX-Turk without ‘and’ and ‘not’ operators and added the results in the revised version.
>
> (3) In the proposed approach, the results NeuralSplitter are final in the sense that after computing the output of the network (splits of the input examples) it cannot be changed and there is no search over these splits. It looks like an error of this stage can be fatal for the overall method. Evaluating how bad this effect is would strengthen the paper.
>
> - In order to evaluate the effect by the results of NeuralSplitter more precisely, we have conducted additional experiments for comparing the regex synthesis performance based on the predictions of the NeuralSplitter with the ground-truth split. We have added the experimental results in the Table 1. in the revised version. It can be seen indirectly what is the upper bound of the performance with new evaluation.
>
> (4) I don't understand why the results of the Blue-Fringe baseline are evaluated so differently w.r.t. the two metrics. Such difference should be easy to see qualitatively on several examples.
>
> - Due to its nature, the Blue-Fringe algorithm often ends up with synthesizing the overfitted regex very quickly compared to the AlphaRegex algorithm. Since the vanilla Blue-Fringe algorithm is very fast, the success rate even decreases when we use the SplitRegex framework because of the runtime overhead caused from the split inference process by the NeuralSplitter model. However, the SplitRegex improves the semantic accuracy of the generated regexes as the process of splitting examples already makes the resulting regexes more fine-grained compared to those generated by the vanilla Blue-Fringe algorithm. We have added further explanation about the disparity of the experimental results with respect to the two metrics in the ‘Main Results’ section.
>
> (5) The overall performance of the SplitRegex (from table 1) is around 60%, which is higher than the baselines, but it still means that there are lots of errors. Showing and analyzing the error cases (on top of Successful examples in Table 3) would greatly strengthen the paper.
>
> - We agree that demonstration and analysis of error cases by the SplitRegex can strengthen the depth of the paper. We will add the error cases in the appendix of the paper by the end of the rebuttal phase.
>
> (6) I also feel like another metric is missing: the percentage of the regexes that are generated correctly in the sense that all the unseen by the generation process positive and negative examples (the ones used in semantic regex accuracy) are recognized correctly within the time limit. This metric is stricter and will indicate how many regexes are actually correctly generated.
>
> - We agree that the suggested metric could be useful to understand the regex synthesis performance of the proposed method. We will add the experimental results regarding the suggested metric to the paper by the end of the rebuttal phase.

---

> > ### Comment · Reviewer_fKY9 · 2021-11-24
> > **comments on the author response**
> >
> > Thanks for your reply. I'll provide more comments below.
> >
> > (1) Please, note that the sole fact that regexes come from some real-world projects does not say much about their complexity. There are many simple regexes in the real-world projects.
> > I also have a feeling that just using the set of excluded regexes is not sufficient to quantify complexity, one should also somehow use the length of some sort.
> >
> > (2) I've looked at the new results in Table 9 and find them suspicious. On both datasets, for all the methods all three metrics give very similar values. This was not the case in other scenarios (in Table 1, the difference is drastic). It would be helpful to explain such behaviour, otherwise it looks like a bug.
> >
> > (3) It looks like the upper bound on performance (with the GT splits) is still quite low (32% for RegExLib and <70% for others), which means that splitting correctly is only a part of the problem, and it is not clear that it is the most significant problem.
> >
> > (4) ok, thanks
> >
> > (5) I'm sorry, I did not find any demonstrations promised.
> >
> > (6) Thanks for adding a new metric, I think it is very useful.
> >
> > (7) I'm still very suprised. I expected to see a baseline that uses a neural-network generative model over regexes as a prior with a search engine on top. It looks like a method like (Ye et al., 2020) would be a good start.
> >
> > (Ye et al., 2020) Optimal Neural Program Synthesis from Multimodal Specifications. Xi Ye, Qiaochu Chen, Isil Dillig, Greg Durrett, arXiv:2010.01678
> >
> > (8) thanks, that works.

---

> > > ### Author Response · Authors · 2021-11-26
> > > **Reply to reviewer fKY9 part 1**
> > >
> > > (1) Please, note that the sole fact that regexes come from some real-world projects does not say much about their complexity. There are many simple regexes in the real-world projects. I also have a feeling that just using the set of excluded regexes is not sufficient to quantify complexity, one should also somehow use the length of some sort.
> > >
> > > -> We might have misunderstood your previous question regarding the complexity of regexes. We first thought that your question is about semantic complexity regexes such as supported operators, not about the lengths of regexes. As we can see in the following paper, the lengths of most practical regexes extracted from real-world software projects are about 15~40 so we think that our approach can be considered ‘useful’ enough if it works for regexes with length in the range. If we can submit the final version of the paper, we will add an experimental result about the performance of our approach for regexes with different length ranges to show how the proposed method works for ‘harder’ cases (possibly for longer regexes).
> > > James C. Davis, Daniel Moyer, Ayaan M. Kazerounl, Dongwoon Lee: Testing Regex Generalizability And Its Implications: A Large-Scale Many-Language Measurement Study. ASE 2019.
> > >
> > >
> > > (2) I've looked at the new results in Table 9 and find them suspicious. On both datasets, for all the methods all three metrics give very similar values. This was not the case in other scenarios (in Table 1, the difference is drastic). It would be helpful to explain such behaviour, otherwise it looks like a bug.
> > >
> > > → The new results in Table 9 could look suspicious but we have double-checked that the result is correct. The main reason is that the size of the test dataset (KB13) is much smaller than the dataset used in Table 1. Note that there are only 441 regexes (after filtering out regexes with unsupported operators) in KB13 while there are 4438 regexes in NL-RX-Turk. Moreover, KB13 and NL-RX-Turk datasets are much simpler (shorter length, less frequent use of the nested operators) than our datasets from Table 1. We include the following regex examples from KB13 to show the simplicity of KB13;
> > >
> > > - agde
> > >
> > > - .*f
> > >
> > > - .*[0-9].*
> > >
> > > - .*q.*r.*
> > >
> > > - .*8.*
> > >
> > > Due to their simplicity, the synthesized regexes tend to accept most of the unseen examples when they succeed in the given examples. This explains why the ‘ratio of fully accurate regexes’ is often equivalent to the ‘success ratio’. Moreover, the ratio of fully accurate regexes is equivalent to the semantic regex accuracy since the semantic regex accuracy is 1 if our method succeeded in regex synthesis and otherwise 0 just like the ratio of fully accurate regexes is. Therefore, the values for all three metrics are the same on KB13.
> > >
> > >
> > > (3) It looks like the upper bound on performance (with the GT splits) is still quite low (32% for RegExLib and <70% for others), which means that splitting correctly is only a part of the problem, and it is not clear that it is the most significant problem.
> > >
> > > -> Considering the fact that the most significant problem in regex synthesis is ‘time efficiency’, we think that the proposed framework could address a very important part of the problem. If we set the timeout value to very high, then the performance will be improved. The main contribution of the paper is to reduce the time for regex synthesis (which takes very long usually) to make the technology more practically useful. We believe that the proposed scheme sufficiently improves the previous baseline algorithms by adopting the ‘divide-and-conquer’ paradigm for the first time. However, we still think that there can be different strategies that can improve the time efficiency of the regex synthesis process as you suggested. We also believe that finding more powerful strategies to the regex synthesis problem is a very interesting future work for us.
> > >
> > >
> > >
> > > (5) I'm sorry, I did not find any demonstrations promised.
> > >
> > > → We have added a demonstration of the error case of SplitRegex in appendix D.6 and Table 10. We understand that the demonstration of only one failure case might not be sufficient. We are planning to add more distinct failure cases (possibly from different reasons) if we can submit the final version.

---

> > > > ### Author Response · Authors · 2021-11-26
> > > > **Reply to reviewer fKY9 part 2**
> > > >
> > > > (7) I'm still very surprised. I expected to see a baseline that uses a neural-network generative model over regexes as a prior with a search engine on top. It looks like a method like (Ye et al., 2020) would be a good start. (Ye et al., 2020) Optimal Neural Program Synthesis from Multimodal Specifications. Xi Ye, Qiaochu Chen, Isil Dillig, Greg Durrett, arXiv:2010.01678
> > > >
> > > > →  Indeed, we were also surprised by the fact that there was no prior research using neural networks for efficient synthesis of regex in the recent literature. We believe this is because using the advantages of neural networks with examples is a very challenging problem. The paper you suggested introduces a multimodal framework that scores the partial program with a neural network based on NL description, and then judges the feasibility of programs via best-first search based on examples. The examples are used only in proving the feasibility of the prioritized program, not as input to the neural network.
> > > > We actually conducted several experiments about neural network generative model over regexes as a prior while utilizing examples as input. However, the experimental results show poorer performance than the considered baselines. The main reason was that the neural network simply failed to understand the relationship between multiple examples and target regex. We still agree that an NN generative model over regexes is a promising direction for future work. We will definitely consider this direction as well in the future.

---

> > > > ### Comment · Reviewer_fKY9 · 2021-11-26
> > > > **comments**
> > > >
> > > > (1) Length of regexes and number of available constructions are two different dimensions to evaluate complexity. It would be nice to somehow deal with both of them.
> > > >
> > > > (2)  > Due to their simplicity, the synthesized regexes tend to accept most of the unseen examples when they succeed in the given examples. This explains why the ‘ratio of fully accurate regexes’ is often equivalent to the ‘success ratio’.
> > > >
> > > > I guess I misunderstood the meaning of the "Full" metric. In my original suggestion, it was supposed to take care of both unseen positives and negatives. Is it true that the current version deals only with unseen positives but not negatives?
> > > >
> > > > (3) I agree that improving the time efficiency of the synthesizers is very important but I have not found detailed experiments on that. All the presented results are obtained for 1 time limit for each method, which is not fully showing the improvement of time efficiency.
> > > >
> > > > (7) Thanks for your reply. I agree that building a baseline/method along the lines of what I mentioned is another research direction. It indeed looks like such a method does not readily exist (which I cannot disprove).

---

> > > > > ### Author Response · Authors · 2021-11-29
> > > > > **Reply to reviewer fKY9**
> > > > >
> > > > > (1) Length of regexes and number of available constructions are two different dimensions to evaluate complexity. It would be nice to somehow deal with both of them.
> > > > > -> Now we fully understand your comment. We will add experimental results on the performance of the proposed method for regexes with different complexities in terms of both lengths of regexes and the number of operations used in regexes.
> > > > >
> > > > > (2) > Due to their simplicity, the synthesized regexes tend to accept most of the unseen examples when they succeed in the given examples. This explains why the ‘ratio of fully accurate regexes’ is often equivalent to the ‘success ratio’.
> > > > > I guess I misunderstood the meaning of the "Full" metric. In my original suggestion, it was supposed to take care of both unseen positives and negatives. Is it true that the current version deals only with unseen positives but not negatives?
> > > > > ->  The ‘Full’ metric is about both unseen positive and negative examples as you suggested. What we meant in the previous response is that the proposed method successfully synthesized regexes that are semantically equivalent to target regexes due to the intrinsic simplicity of problems in KB13. Since the synthesized regexes are semantically equivalent (language-equivalent) to target regexes, they often satisfy all unseen positive and negative examples.
> > > > >
> > > > > (3) I agree that improving the time efficiency of the synthesizers is very important but I have not found detailed experiments on that. All the presented results are obtained for 1 time limit for each method, which is not fully showing the improvement of time efficiency.
> > > > > -> Please find Table 6 in the appendix section which is about the regex synthesis performance of the proposed model and baselines under different timeout values. We tested with three time limits (1, 2, and 3 seconds) and demonstrated the time efficiency of the proposed framework. Moreover, we also have directly compared the average runtime taken for regex synthesis of the vanilla AlphaRegex with the AlphaRegex with the SplitRegex framework. This result shows that the proposed framework improves the time efficiency of regex synthesis in almost 79.18% of test cases compared to the vanilla AlphaRegex and the success ratio to 46.10% from 25.77%. The average runtime reduces to 1.6622 (0.0088 for successful cases) from 2.4344 (0.1250 for successful cases) if we apply the SplitRegex to the vanilla AlphaRegex.
> > > > >
> > > > > (7) Thanks for your reply. I agree that building a baseline/method along the lines of what I mentioned is another research direction. It indeed looks like such a method does not readily exist (which I cannot disprove).
> > > > > -> Thank you very much for your valuable suggestion.

---

> ### Author Response · Authors · 2021-11-21
> **Response to reviewer fKY9. Part 2**
>
> (7) It looks like none of the baselines uses a neural network. Is it actually true that no published work put positive/negative regex examples into a neural network?
>
> We acknowledge that some studies aim to extract a finite automata from a trained neural network, However, there are no published papers that infer a regular expression directly from the pos/neg examples as far as we are aware.
> C. Jiang et al., Cold-start and interpretability: Turning regular expressions into trainable recurrent neural networks, EMNLP 2020 (https://aclanthology.org/2020.emnlp-main.258.pdf)
> T. Okudono et al., Weighted automata extraction from recurrent neural networks via regression on state spaces, AAAI-20 (https://ojs.aaai.org/index.php/AAAI/article/view/5977/5833)
> G. Weiss et al., Extracting automata from recurrent neural networks using queries and counterexamples, ICML 2018 (https://proceedings.mlr.press/v80/weiss18a/weiss18a.pdf)
>
> (8) I did not find a description of how numerical targets of NeuralSplitter (like 1223344) were defined. I have my guesses but am not sure.
>
> In the revised version of the manuscript, we have added a more detailed description about how numerical targets of NeuralSplitter are defined at the beginning of ‘Our Approach’ section.

---

### Official Review · Reviewer_Nx3i · 2021-11-03

**Correctness:** 3
**Technical Novelty And Significance:** 2
**Empirical Novelty And Significance:** 2
**Recommendation:** 5
**Confidence:** 4

**Main Review:**

*Strengths*
- The motivation of the work is well laid out.

*Weaknesses*
 - Lack of clarity: I found many important details from the paper missing. Some of them include:
 (i) What is L ? It is stated directly in Section 3 without defining what it is and then used consistently there after without any clarification.
 (ii) Why is there a disparity between the performance of SplitRegex in terms of the two metrics (success rate and semantic regex accuracy)? From Table 1 we can see that across all datasets, SplitRegex is better only in terms of semantic regex similarity but not the success rate. Moreover, I would have expected the combination of SplitRegex with Blue-Fringe to be better in terms of semantic similarity keeping in view that without SplitRegex, Blue-Fringe is the best framework. This disparity is worrying for me and understanding why this occurs is required to put more faith in the SplitRegex system.
(iii) There are some questionable design choices made by the authors elucidating in Section 3.1 as well as the treatment of the wildcard pattern. It seems to me that these decisions were made to fit the SplitRegex framework in particular without proper motivations justifying why these choices make sense on a system design level.
(iv) It is not clear to me why SplitRegex works better with an increase in alphabet size while other approaches have the opposite behaviour.
(v) Though the paper claims in the conclusions that their framework is " ...very effective in reducing the time complexity of regex synthesis", this has not been shown experimentally.

-  Weak experimental results: To me, the poor performance of SplitRegex in terms of success rate is concerning ( see point (ii) above)
-  Missing related works: There are very closely related works based on the idea of divide and conquer in program synthesis that are missing from the paper. Some include:

   [1] Shrivastava, Disha, H. Larochelle and Daniel Tarlow. “Learning to Combine Per-Example Solutions for Neural Program Synthesis.” NeurIPS, 2021. They find solutions that satisfy a single example (split) and then learn to combine the solutions (combine) using a neural network.

   [2] Shraddha Barke, Hila Peleg, and Nadia Polikarpova. 2020. Just-in-time learning for bottom-up enumerative synthesis. Proc. ACM Program. Lang. 4, OOPSLA, Article 227 (November 2020), 29 pages. DOI:https://doi.org/10.1145/3428295

- Less novel: In lieu of the other works mentioned above that have already explored the idea of example splitting and then combining the solutions for neural program synthesis, the contributions of the paper seem incremental.

-  Narrow scope: It is not exactly clear to me how the approach will be used for other domains say string manipulations programs or straight-line code or programs containing data and control-flow structures. It seems the approach is catered specifically to regular expressions that make the approach extremely narrow in scope and hence less significant.

*Suggestions/Comments*
- Figure 1 and Figure 2 can be combined into a single Figure. Both have a lot of redundant information.
- Results on other values of timeout and other domains ( apart from Regex) will be useful.

**Summary Of The Paper:**

The paper proposes a method to synthesize regular expressions by first splitting positive examples and synthesizing regexes for those and then combining these while using negative examples as an additional constraint. They perform experiments on few regex baselines and show better results in one of the two metrices.

**Summary Of The Review:**

The paper has less technical novelty, weak experimental results, lack of clarity, missing related works and narrow scope in terms of applicability to other programs that are not regex. I will suggest improvement in writing in terms of clarity and a thorough evaluation across other DSLs.

---

> ### Author Response · Authors · 2021-11-21
> **Response to reviewer Nx3i. Part 1**
>
> We thank the reviewer for the valuable comments and detailed suggestions. We respond to the reviewer’s comments as follows.
>
> (1) I found many important details from the paper missing. Some of them include: (i) What is L ? It is stated directly in Section 3 without defining what it is and then used consistently there after without any clarification.
>
> - We have add the definition of $L_p$ in Section 3 as suggested in the revised version.
>
> (2) Why is there a disparity between the performance of SplitRegex in terms of the two metrics (success rate and semantic regex accuracy)? From Table 1 we can see that across all datasets, SplitRegex is better only in terms of semantic regex similarity but not the success rate.
>
> - The success rate of the Blue-Fringe algorithm is the highest among all the considered algorithms due to the nature of the Blue-Fringe algorithm. The Blue-Fringe algorithm merges the suffixes of given positive examples while assuming that any string the expression represents is positive unless there is a counterexample---negative examples.   While the Blue-Fringe algorithm can simply generate a regex satisfying the examples very quickly, the generated regex represents a slightly larger set of strings than the set of positive examples in general. This means that the regex generated by Blue-Fringe rarely accepts positive examples that are not used in the regex synthesis process. We have added this part in the ‘Detailed Analysis’ section of the revised version.
>
> (3) Moreover, I would have expected the combination of SplitRegex with Blue-Fringe to be better in terms of semantic similarity keeping in view that without SplitRegex, Blue-Fringe is the best framework. This disparity is worrying for me and understanding why this occurs is required to put more faith in the SplitRegex system.
>
> - As we explained above, the semantic accuracy is a more important metric for evaluating the quality of the generated regex. We would like to emphasize the point that the SplitRegex framework improves the semantic accuracy of Blue-Fringe while the success rate slightly decreases due to the runtime overhead caused by the inference of the regex split.
>
> (4) There are some questionable design choices made by the authors elucidating in Section 3.1 as well as the treatment of the wildcard pattern. It seems to me that these decisions were made to fit the SplitRegex framework in particular without proper motivations justifying why these choices make sense on a system design level.
>
> - We exclude regexes with backreference, lookahead/lookbehind, and numerical quantifiers because all of the baseline algorithms such as AlphaRegex, Blue-Fringe, and RegexGenerator++ are not able to synthesize regexes with the features. Since our SplitRegex framework works on top of the existing regex synthesizers, we should have selected regexes that can be synthesized by the existing regex synthesizers.
>
> (5) It is not clear to me why SplitRegex works better with an increase in alphabet size while other approaches have the opposite behaviour.
>
> - The AlphaRegex algorithm relies on the bottom-up best-first search that enumerates from the most coarse regex .* to more fine-grained regexes by replacing each terminal node in regex parse tree with more fine-grained regex parse trees. Therefore, the performance of AlphaRegex naturally gets worse as the alphabet size increases since the search space also increases exponentially. Roughly speaking, the complexity of search space is in O(m^n) when m is the alphabet size and n is the length of regexes. Meanwhile, the proposed SplitRegex works better as the alphabet size increases as the task of splitting regex is easier with larger alphabet size. The NeuralSplitter model can easily predict the split just by grouping the substrings with similar sets of symbols when the alphabet size is larger. As a result, the performance of AlphaRegex can be improved when used with SplitRegex since the search space complexity decreases from O(m^n) to O(m^{n/s}) when s is the split size.
>
> (6) Though the paper claims in the conclusions that their framework is " ...very effective in reducing the time complexity of regex synthesis", this has not been shown experimentally.
>
> - We have shown the experimental results supporting the claim that our framework is very effective in reducing the time complexity of regex synthesis by Table 7 in the appendix.
>
> (7) (On Weaknesses #2 (Weak experimental results)) To me, the poor performance of SplitRegex in terms of success rate is concerning ( see point (ii) above)
>
> - We have answered the comments in response to the comment (2) above.

---

> ### Author Response · Authors · 2021-11-21
> **Response to reviewer Nx3i. Part 2**
>
> (8) (On Weaknesses #3 (Missing related works)) There are very closely related works based on the idea of divide and conquer in program synthesis that are missing from the paper. Some include:
>
> - We have added the suggested related works as references.
>
> (9) (On Weaknesses #4 (Less novel)) In lieu of the other works mentioned above that have already explored the idea of example splitting and then combining the solutions for neural program synthesis, the contributions of the paper seem incremental.
>
> - We agree that the suggested related works have explored the idea of example splitting but their approaches are substantially different from our approach as they simply build solutions per each example and combine the solutions to find the global solution while we train a neural network for splitting given examples. When there are five examples, they build a solution for each example and combine them to build the global solution. They do not consider ‘how’ to split examples and just consider each example at a time separately. On the other hand, we try to find subproblems from the five examples by grouping the similar substrings from all of the five examples. Literally, our neural network should learn ‘how to split’ the given examples so that we can solve easier subproblems from successfully reduced search space.
>
> (10) (On Weaknesses #5 (Narrow scope)) It is not exactly clear to me how the approach will be used for other domains say string manipulations programs or straight-line code or programs containing data and control-flow structures. It seems the approach is catered specifically to regular expressions that make the approach extremely narrow in scope and hence less significant.
>
> - We agree that it might broaden the scope of the current paper if we can apply a similar approach to other domains such as string manipulations or straight-line programs. However, we think that the regex synthesis problem is a very challenging problem in computer science as it is. For example, regexes are important in text/string processing such as detecting malicious patterns in network streams. These problems need an appropriate regular expression or they have a problem, named ReDoS (regular expression DoS) that harms systems. Actually, Cloudflare had an outage in 2019 due to this specific problem. Therefore a proper inference scheme for regular expressions is important not only in the classical sense but in several application fields. We can also easily find many fruitful research outcomes in recent years only focused on the regex synthesis problem as follows:
>
> Yeting Li, Shuaimin Li, Zhiwu Xu, Jialun Cao, Zixuan Chen, Yun Hu, Haiming Chen, Shing-Chi Cheung: TRANSREGEX: Multi-modal Regular Expression Synthesis by Generate-and-Repair. ICSE 2021.
>
> Zexuan Zhong, Jiaqi Guo, Wei Yang, Jian Peng, Tao Xie, Jian-Guang Lou, Ting Liu, Dongmei Zhang: SemRegex: A Semantics-Based Approach for Generating Regular Expressions from Natural Language Specifications. EMNLP 2018
>
> Alberto Bartoli, Andrea De Lorenzo, Eric Medvet, Fabiano Tarlao: Inference of Regular Expressions for Text Extraction from Examples. IEEE Trans. Knowl. Data Eng 2016.
>
> Nicholas Locascio, Karthik Narasimhan, Eduardo DeLeon, Nate Kushman, Regina Barzilay: Neural Generation of Regular Expressions from Natural Language with Minimal Domain Knowledge. EMNLP 2016.
>
> Yeting Li, Zhiwu Xu, Jialun Cao, Haiming Chen, Tingjian Ge, Shing-Chi Cheung, Haoren Zhao: FlashRegex: Deducing Anti-ReDoS Regexes from Examples. ASE 2020.
>
> Definitely, we will continue the line of research by applying the divide-and-conquer paradigm in various program synthesis problems including string manipulations and straight-line programs as suggested.
>
>
> (11) (Suggestions/Comments #1 (combine Figures) Figure 1 and Figure 2 can be combined into a single Figure. Both have a lot of redundant information.
>
> - We are grateful for the valuable suggestion. We will combine the figures into a single figure in the revised version.
>
> (12) (Suggestions/Comments #2 (Additional results) Results on other values of timeout and other domains ( apart from Regex) will be useful.
>
> - We have added the results on other values of timeout (5 and 10 seconds) in the appendix due to the page limit of the main text. Regarding the results on other domains, we would like to borrow the answer to the comment (10).

---

### Official Review · Reviewer_Nyta · 2021-11-03

**Correctness:** 4
**Technical Novelty And Significance:** 2
**Empirical Novelty And Significance:** 3
**Recommendation:** 6
**Confidence:** 3

**Main Review:**

Overall the idea is neat and simple, and the consistent improvement from the existing toolkits is appealing. I have several comments below:

1. The modeling part seems a bit arbitrary. While I understand that this paper is a kind of proof-of-concept, the segmentation model can potentially be improved in the following ways:
- The RNN mentioned in appendix A is unidirectional. However one may want to at least use bidirectional ones in Sec 3.2, for RNN_enc1 (or Transformers)
Ideally the component of RNN_enc2 should be order invariant (i.e., encode a set of strings, rather than a list of strings with particular order), something like DeepSet should be considered
- Ideally the predicted segmental label should be permutation invariant as well.
- Regarding the evaluation of the segmentation model itself, one can use the optimal transport metrics, following the practice in the literature.
I would suggest the authors investigate a bit more carefully into the literature of segmentation models, either supervised or unsupervised.

2. Is it possible to evaluate the performance based on the ground-truth split, rather than the model predicted splits? In this way we can get to know what is the upper bound of the performance, and see how far away we are from this upper bound.

3. The model relies on the existing synthesis tools for the sub-RegExpr synthesis, so the bottleneck would be the synthesis of the most difficult sub-expression. Imagine if the sequence is formed from a single segment and in this case the proposed model would not help in any way. I’m wondering how often this happens, or if there’s a dominant segment that takes the most amount of time, how much speed up can we get with the neural split model?


**Summary Of The Paper:**

This paper proposed a way of synthesizing RegEx from positive and negative sample specifications. The main idea is to split the positive samples into segments, such that one can leverage any existing RegEx synthesis tool to synthesize the sub-regex out of the segments of specifications. Finally the entire RegEx is obtained via the concatenation of these sub-RegExes. The paper proposes to split the specifications using RNN, which is trained via synthetically generated data. Experiments on synthetic and some benchmark datasets show that the proposed method is able to outperform the existing counterparts in most cases.

**Summary Of The Review:**

The paper presents a practical and neat idea for RegEx synthesis that can leverage existing synthesis tools as subroutines, based on the proposed neural-split model. However the neural splitter is not technically solid, and there can be more experimental studies for the sake of completeness and the understanding of current limitations.

---

> ### Author Response · Authors · 2021-11-21
> **Response to reviewer Nyta**
>
> We thank the reviewer for the valuable comments and detailed suggestions. We respond to the reviewer’s comments as follows.
>
> (1) The RNN mentioned in appendix A is unidirectional. However one may want to at least use bidirectional ones in Sec 3.2, for RNN_enc1 (or Transformers) Ideally the component of RNN_enc2 should be order invariant (i.e., encode a set of strings, rather than a list of strings with particular order), something like DeepSet should be considered.
>
> - RNN_encoders used in our neural network are actually bi-directional. We have briefly mentioned it in the ‘hyperparameter’ paragraph in the original version as follows: “We use two-layer bidirectional GRU with 256 hidden units unless explicitly mentioned otherwise”.
> We have added explanations about the bi-directionality of the RNN_encoders in the `Neural Network Architecture’ section to further clarify it in the revised version.
>
> (2) Ideally the predicted segmental label should be permutation invariant as well.
>
> - Ideally, maybe we can make the predicted label to be permutation invariant. For instance, maybe we can encode ‘112233’ as ‘101010’, where ‘1’ indicates the beginning of new split. However, there might be input positive strings with labels such as ‘1133’ and ‘2244’ that do not contain substrings corresponding to some splits. For this reason, we use the current encoding scheme which is not permutation invariant.
>
> (3) Regarding the evaluation of the segmentation model itself, one can use the optimal transport metrics, following the practice in the literature. I would suggest the authors investigate a bit more carefully into the literature of segmentation models, either supervised or unsupervised.
>
> - We are grateful for the valuable suggestion. We will investigate more into the other segmentation models and try to incorporate the insights into our future work to improve the result.
>
> (4) Is it possible to evaluate the performance based on the ground-truth split, rather than the model predicted splits? In this way we can get to know what is the upper bound of the performance, and see how far away we are from this upper bound.
>
> - We agree that it is a very good idea to evaluate the regex synthesis performance based on the ground-truth split. We are now conducting experiments as suggested and planning to add the experimental results by the end of the rebuttal phase.
>
> (5) The model relies on the existing synthesis tools for the sub-RegExpr synthesis, so the bottleneck would be the synthesis of the most difficult sub-expression. Imagine if the sequence is formed from a single segment and in this case the proposed model would not help in any way.
>
> - We examined the number of regexes with split size 1 from all used datasets and found that only 0%, 6.07%, and 2.57% of regexes from Snort, RegexLib, and Polyglot datasets, respectively, have split size 1. Note that we randomly sampled 1,000 regexes from each dataset. So we can claim that most practical regexes consist of more than one segment and our model could help in most practical cases.
>
> (6) I’m wondering how often this happens, or if there’s a dominant segment that takes the most amount of time, how much speed up can we get with the neural split model?
>
> - For practical regexes consisting of a single ‘segment’, the NeuralSplitter predicted that 73.02% (from RegexLib) and 75.00% (from Polyglot) of regexes have split size larger than one. Interestingly, our method (SplitRegex + AlphaRegex) shows similar performance to the baseline (AlphaRegex) even for these regexes. While the baseline succeeded in 15% and 43% in RegexLib and Polyglot (there was no regex with split size 1 in Snort), our method succeded in 20% and 38% of the regexes.

---

### Official Review · Reviewer_qHid · 2021-11-04

**Correctness:** 4
**Technical Novelty And Significance:** 3
**Empirical Novelty And Significance:** 4
**Recommendation:** 8
**Confidence:** 3

**Main Review:**

Strengths:
- this is a novel application for recurrent neural networks, and I am excited to see such innovative uses of neural heuristic solutions for otherwise unsolvable, hard problems
- the experiments section is solid, and presents a few different baselines and benchmark datasets, both including real, practical examples, and synthetically generated ones.
- the authors demonstrate and motivate the benefit that SplitRegex adds when applied in combination with existing regular expression generators
- the paper is well-written and is easy to read and follow

Weaknesses:
- My main concern is about the approach for generating negative strings. For example, the authors do not specify whether they make sure that their negative examples are *not* matched by their regular expression. I also have doubts whether substituting a random symbol in a string really produces a very challenging counter-example; I think this is further illustrated in the Figure 3, where the performance of the model goes up as the size of the alphabet increases - because the task gets easier. The authors have presented a "hard" negative example "aabbccc", which can be incorrectly parsed by a regular expression that is too generic, but generating this "hard" negative from positive example "aabbcc" is less likely if the size of the alphabet is larger. One thing I would probably suggest to do instead could be introducing the noise at the level of regular expression itself, and then sampling strings from the "negative or incorrect" regular expression, but this might not be a catch-all solution either, besides the fact that deciding whether two regexes are the same or not is not trivial. I definitely think that further experimentation in this area would improve the value of the paper.

- I also have a reservation about the 3 seconds time limit, as it seems to be chosen arbitrarily. I understand that it might not be practical to experiment with a larger time limit for a really large, randomly generated dataset, but at least attempting evaluating some of the smaller datasets would be great. and I am also not convinced that it is chosen wisely from the empirical perspective, because one of the baselines does not find *any* regular expressions in that amount of time, which makes including it less valuable and less informative for the reader.

- the paper could be improved via further editing and proof-reading, some of the smaller typos I noticed:
> "do not produce desired regexes within a give time limit." -> "a given"
> "For each regex, we randomly generate 20 positive and negative examples of length up to certain length"
> "Recurrent network networks"

**Summary Of The Paper:**

This paper proposes a novel method for employing neural networks to aid in solving an NP-hard problem. The authors use the divide-and-conquer approach to regular expression generation, by generating the regex one piece at a time. They use Recurrent Neural Network to predict which substrings should be tackled by the same sub-regular expression, and then combine multiple subregular expressions into a single regular expression.
The authors demonstrate how their proposed approach is helpful in combination with different regular expression generation tools, and demonstrate improved overall performance in all cases. In the experiments where SplitRegex algorithm is applied in combination with a fast regex generator (BF), their experiments indicate that some of the speed is getting traded off for semantic accuracy. In the experiments with the slowest regex generator (RegexGenerator++), they manage to speed up its computation time to go from timing out before producing any regexes to successfully producing regexes in 20% of the time. Finally, in the experiments with AlphaRegex they demonstrate that both success rate and the accuracy go up on all benchmark datasets.

**Summary Of The Review:**

This is a novel approach to a problem that does not have feasible exact solutions, and I think it is an interesting use of neural networks. The proposed approach is relatively light-weight and demonstrates improvement in both run-time and performance quality in most cases. It has been evaluated quite rigorously, further evaluation can improve the quality of the paper, but even as it is, the proposed approach demonstrates non-trivial improvement for real-world scenario datasets, which I believe warrants acceptance of this paper.

---

> ### Author Response · Authors · 2021-11-21
> **Response to reviewer qHid**
>
> We thank the reviewer for the valuable comments and detailed suggestions. We respond to the reviewer’s comments as follows.
>
> (1) “My main concern is about the approach for generating negative strings. For example, the authors do not specify whether they make sure that their negative examples are *not* matched by their regular expression. “
>
> - When generating negative samples, we checked that each negative sample is not matched by the corresponding regex. We have explicitly mentioned this condition and also elaborated our approach for generating negative examples in the revised version.
>
> (2) I also have doubts whether substituting a random symbol in a string really produces a very challenging counter-example; I think this is further illustrated in the Figure 3, where the performance of the model goes up as the size of the alphabet increases - because the task gets easier. The authors have presented a "hard" negative example "aabbccc", which can be incorrectly parsed by a regular expression that is too generic, but generating this "hard" negative from positive example "aabbcc" is less likely if the size of the alphabet is larger.
>
> - We agree that the task gets easier as the size of the alphabet increases and generating challenging negative examples is a very important problem. However, we do not think that it is due to the substitution of a random symbol.The task gets easier with a larger-size alphabet because it is easier to group the similar substrings using common symbols as there are more symbols in a string. We also think that our method is able to generate challenging negative examples as they should be very similar to positive examples and different only at a single symbol. Please note that substituting a random symbol in a string produces more challenging negative examples for the regex learning problem---the model should be able to distinguish a subtle (such as symbol substitution) change between a positive and a negative string. If negative strings are very different, the model tends to produce a union of all positive samples, which is a correct answer, and thus, the problem becomes trivial instead of being challenging.
>
> (3) One thing I would probably suggest to do instead could be introducing the noise at the level of regular expression itself, and then sampling strings from the "negative or incorrect" regular expression, but this might not be a catch-all solution either, besides the fact that deciding whether two regexes are the same or not is not trivial. I definitely think that further experimentation in this area would improve the value of the paper.
>
> - We agree with your suggestion about ‘introducing the noise at the level of regex’ is reasonable. We have conducted experiments for the case and added the experimental results in the appendix D.4 in the revised version. We observe that the dataset noised at the level of regex generates harder negative examples than the previous one.
>
> (4) I also have a reservation about the 3 seconds time limit, as it seems to be chosen arbitrarily. I understand that it might not be practical to experiment with a larger time limit for a really large, randomly generated dataset, but at least attempting evaluating some of the smaller datasets would be great. I am also not convinced that it is chosen wisely from the empirical perspective, because one of the baselines does not find any regular expressions in that amount of time, which makes including it less valuable and less informative for the reader.
>
> - First, we already had conducted experiments with 1, 2, and 3 seconds time limits in the appendix. Now we have conducted additional experiments with larger time limits (5 and 10 seconds) as you suggested and added the experimental results in the revised version.
>
> (5) The paper could be improved via further editing and proof-reading, some of the smaller typos I noticed...
>
> - We are very grateful to the reviewer for noticing the typos. We have revised the typos as you suggested.

---

### Decision · Program_Chairs · 2022-01-20

**Decision:**

Reject

**Comment:**

There were genuine differences of opinion here. I saw reviews of 8,6,5,5.
In these cases, I do try to check if the 8 has a really compelling argument and err on the side of accepting, but here I think both the positive and negative reviews have fair points, so I am inclined to recommend rejection here.

I think the good news is that a lot of the negative stuff was around scoping/writing/related-work, and so it should be (relatively) easy to shore up this submission into something that will get better reviews in the next conference cycle.